# Exposure–Response Relationships for Toceranib in Dogs with Solid Tumors: A Pilot Study

**DOI:** 10.3390/ani15071025

**Published:** 2025-04-02

**Authors:** Young-Rok Kim, Ji-Hwan Park, Kieun Bae, Kyong-Ah Yoon, Jung-Hyun Kim

**Affiliations:** 1KU Animal Cancer Center, Konkuk University Veterinary Medical Teaching Hospital, Seoul 05029, Republic of Korea; 2Department of Veterinary Internal Medicine, College of Veterinary Medicine, Konkuk University, Seoul 05029, Republic of Korea; 3Bundang Leaders Animal Medical Center, Seongnam 13496, Republic of Korea; 4Department of Veterinary Biochemistry, College of Veterinary Medicine, Konkuk University, Seoul 05029, Republic of Korea

**Keywords:** dog, exposure–response relationship, oncology, solid tumor, toceranib phosphate, targeted therapy

## Abstract

Toceranib phosphate, the most commonly used tyrosine kinase inhibitor in veterinary oncology, meets most of the criteria for therapeutic drug monitoring (TDM); however, its exposure–response relationships have not been established. Therefore, we evaluated correlations between plasma toceranib phosphate concentrations and its efficacy and safety in dogs with solid tumors and explored its feasibility for TDM. In this study, we included 10 dogs with seven solid tumors, including hepatocellular carcinoma (*n* = 4), squamous cell carcinoma (*n* = 1), soft tissue sarcoma (*n* = 1), histiocytic sarcoma (*n* = 1), oral malignant melanoma (*n* = 1), mammary carcinoma (*n* = 1), and pulmonary carcinoma (*n* = 1). A steady-state plasma concentration was achieved within 1 week. A higher steady-state peak plasma concentration may be associated with an increased risk of adverse events. Although large-scale studies are needed, our findings indicate that toceranib phosphate is a suitable candidate for TDM and suggest that adjusting its dose based on drug exposure could prevent toxicity.

## 1. Introduction

Toceranib phosphate (TOC), an orally available small-molecule tyrosine kinase inhibitor (TKI), is the only targeted therapy for dogs approved by the Food and Drug Administration [1]. As a multitarget inhibitor, TOC competitively inhibits the binding of adenosine triphosphate to various receptor tyrosine kinases (RTKs) in the split-kinase family, including vascular endothelial growth factor receptor 2, platelet-derived growth factor receptor α/β, and stem cell factor receptor [1]. Dysregulation of these RTKs is linked to the development of cancer [2], and TOC exerts antitumor and antiangiogenic effects by blocking RTK-mediated signaling pathways [1,3,4].

TOC was originally approved for the treatment of dogs with advanced mast cell tumors [5]. TOC is also effective against various types of canine solid tumors, with a clinical benefit of 50–90% [1,6,7]. In the adjuvant setting, TOC can prevent local recurrence or distant metastasis by modulating the tumor microenvironment [8,9].

In a phase 1 dose escalation study, the starting dose of TOC of 3.25 mg/kg every other day was determined to be the maximum tolerated dose (MTD); however, the biological activity in 11/16 dogs (69%) treated with 2.5 mg/kg TOC every other day was comparable to that in 12/20 dogs (60%) treated with TOC at the MTD [1]. These findings are consistent with the data from a retrospective study showing that TOC exhibits significant biological activity at a median dose of 2.8 mg/kg [6]. Furthermore, treatment at lower doses results in a reduced adverse event (AE) profile and fewer changes in the drug regimen than those at the MTD [6,10]. These data demonstrate that even at doses below the MTD, TOC may remain effective in mitigating toxicity.

In human medicine, owing to interpatient variability in pharmacokinetics (PK) and the relationships between exposure and clinical efficacy and safety, therapeutic drug monitoring (TDM) has been proposed to individualize the dosage of various TKIs [11,12]. For sunitinib, a structural analog of TOC, dose individualization has been proposed for patients with metastatic renal cell tumors, owing to its large interpatient PK variability (34–60%), and TDM-guided dose modification of sunitinib has achieved better outcomes while preventing toxicity [13,14]. Similarly, for TOC, large interpatient variability in plasma exposure was observed in a previous PK study, with interpatient coefficients of variation of 35–40% for C_max_ and 30–61% for C_min_ [15]. However, the exposure–response relationship for TOC in dogs has not been clearly established. C_max_ levels of at least 40 ng/mL appear to be associated with clinical efficacy in dogs with mast cell tumors receiving TOC at the MTD [15]. Additionally, even when TOC was administered within a dosage range of 2.4 to 2.9 mg/kg, the resulting C_max_ exceeded the threshold of 40 ng/mL, and TOC exhibited a reduced AE profile [10].

Given this context, the purpose of this study was to evaluate correlations between TOC exposure and its clinical efficacy and safety and to explore the feasibility of TDM for optimizing TOC treatment in dogs with solid tumors.

## 2. Materials and Methods

### 2.1. Animals

A prospective, randomized clinical trial was conducted on client-owned dogs diagnosed with malignant solid tumors between May 2022 and October 2023 at the Konkuk University Animal Cancer Center. TOC (Palladia^®^, Zoetis, Parsippany, NJ, USA) was administered orally at a dose range of 2.4–2.9 mg/kg every other day with or without food [10]. Medical data for the dogs were collected, including signalment (age, sex, and neuter status); physical examination, blood pressure, hematology, biochemistry, urinalysis, diagnostic imaging (radiography, ultrasound, and computed tomography), and histopathological findings (surgical margin and mitotic count); TOC dose and duration of TOC treatment; and AEs and clinical outcomes. Dogs were excluded from this study if distant metastases were confirmed at the time of diagnosis or if the TOC treatment duration was less than 1 month, which is insufficient for evaluating its effect [16]. This clinical trial was approved by the Institutional Animal Care and Use Committee of the Konkuk University (approval no. KU 23046).

### 2.2. Assessment of Clinical Outcomes and AEs

Rechecks were performed every 1–2 months after the initiation of TOC treatment through physical examination, three-view thoracic radiography, and abdominal ultrasonography.

AEs were evaluated before TOC treatment (at baseline) and 1, 2, and 4 weeks thereafter, followed by regular check-ups every 1–2 months. At each follow-up, a physical examination, blood pressure measurement, hematological and biochemical tests, and urinalysis were performed. AEs were graded according to the Veterinary Cooperative Oncology Group Common Terminology Criteria for AEs (v2.0) [17]. Concomitant medications, including famotidine, omeprazole, silymarin, ursodeoxycholic acid, mirtazapine, and tramadol, were allowed to prevent or treat drug-related AEs. TOC was permanently discontinued upon request of the owner or if unacceptable AEs occurred or disease progression was identified.

### 2.3. Blood Sampling

Whole blood (1–2 mL) was obtained from the external jugular vein after initiation of postoperative TOC treatment. Blood samples were collected in ethylenediaminetetraacetic acid tubes 6 h (C_max_) and 48 h (C_min_) after administration; these time points were determined based on previous studies [10,14]. Within 30 min of collection, the samples were centrifuged at 3000× *g* for 10 min at 4 °C. The plasma was separated, divided into cryovials, and stored at −80 °C until analysis. C_max_ was measured at weeks 1, 4, and 12 of TOC treatment, and C_min_ was measured once during treatment.

### 2.4. Measurement of Plasma Toceranib Concentration

Plasma toceranib was quantitated via high-performance liquid chromatography with tandem mass spectrometry detection (LC–MS/MS). A 100 μL aliquot of calibration standards (5, 10, 20, 50, 100, 200, and 500 ng/mL), quality control samples (15, 40, and 400 ng/mL), and study samples were mixed with 400 μL of an internal standard solution (toceranib-d8, 10 ng/mL in 0.1% formic acid in methanol). After thorough vortexing for 1 min, the mixture was centrifuged at 15,000× *g* for 10 min at 4 °C, and 2 μL of the supernatant was collected and injected into the LC–MS/MS system.

The LC–MS/MS system comprised an ExionLC™ AD system (Sciex, Toronto, ON, Canada) coupled with a Triple Quad™ 5500+ mass system (Sciex). Data were acquired using Analyst 1.4.3. Chromatographic separation was performed using an XBridge C18 column (100 × 2.1 mm, 5 μm; Waters Corporation, Milford, MA, USA) maintained at 40 °C. An isocratic mobile phase comprising 0.1% formic acid in water and 0.1% formic acid in acetonitrile (30:70, *v*/*v*) was used at a flow rate 0.50 mL/min, with the total run time not exceeding 3 min. The autosampler was maintained at 15 °C. All measurements were conducted using a mass spectrometer operated in the positive electrospray ionization mode. The multiple reaction monitoring transitions were *m*/*z* 397.2 → 283.0 for toceranib and *m*/*z* 405.2 → 283.1 for the toceranib-d8 internal standard (IS). The other parameters were as follows: collision gas, curtain gas, ion source gas 1, and ion source gas 2 at 7, 35, 50, and 70 psi, respectively; dwell time, 200 ms; ion spray voltage and temperature, 4000 V and 600 °C, respectively; declustering potential, 80 V for toceranib and the IS; collision energy, 40 V for toceranib and the IS; entrance potential (EP), 12 V for toceranib and 8 V for IS; and collision exit potential, 16 V for toceranib and 18 V for the IS.

Standard curves were generated using the peak area ratios of toceranib to the IS and 1/x^2^ weighted linear regression. Toceranib concentrations were determined via interpolation across a range of 5–500 ng/mL.

### 2.5. Statistical Analysis

Continuous variables are expressed as the mean ± standard deviation. Interpatient variability is expressed as the coefficient of variation, calculated as standard deviation divided by the mean and expressed as a percentage. Normality was assessed using the Shapiro–Wilk test. Continuous variables were compared using Student’s *t*-test or Mann–Whitney *U* test. Linear mixed models for repeated measures were used to compare plasma TOC concentrations over time. Data were analyzed using SPSS v26.0 (IBM Corp., Armonk, NY, USA). Differences were considered significant at *p* < 0.05.

## 3. Results

### 3.1. Patient Characteristics

Ten dogs with histologically confirmed malignant solid tumors were enrolled, with a median age of 10 years (range, 5–12) and median body weight of 6.7 kg (range, 2.3–14.8). Ten breeds were represented: mixed, Pomeranian, Bichon Frisé, Boston terrier, Chihuahua, Coton de Tulear, Japanese spitz, Poodle, Scottish terrier, and Yorkshire terrier. Tumor types included hepatocellular carcinoma (*n* = 4), squamous cell carcinoma (*n* = 1), soft tissue sarcoma (*n* = 1), histiocytic sarcoma (*n* = 1), oral malignant melanoma (*n* = 1), mammary carcinoma (*n* = 1), and pulmonary carcinoma (*n* = 1). Patient characteristics are presented in Table 1. All dogs underwent surgical excision of the primary tumor. The median time interval from surgery to postoperative TOC treatment was 29 d (range, 14–47 d).

### 3.2. Analysis of Plasma Toceranib Concentration

The median dose of toceranib was 2.58 mg/kg (range, 2.07–2.78). None of the dogs underwent dose modification during the study period. During the follow-up period, three out of ten dogs (30%) developed local recurrence and five out of ten (50%) experienced AEs (Appendix A). Measurements of C_max_ and C_min_ of TOC were available for nine and four dogs, respectively. Detailed information for each case is presented in Table 2.

To compare C_max_ according to the time of sample collection, values were normalized to a dose of 3.25 mg/kg, as dose proportionality was observed in a previous PK study [14]. The mean interpatient variabilities in dose-normalized C_max_ and C_min_ were 29% and 61%, respectively. The mean dose-normalized C_max_ values at weeks 1, 4, and 12 were 95.61 ± 31.21, 102.42 ± 34.85, and 82.36 ± 15.93 ng/mL, respectively. No significant differences were noted among mean dose-normalized C_max_ values at weeks 1, 4, and 12 (*p* = 0.414) (Figure 1).

For the exposure–efficacy analysis, C_max_ values at week 1 and on average were used to compare dogs with and without tumor recurrence. Of the nine dogs, three (33.3%) developed tumor recurrence. The overall mean C_max_ values at week 1 and on average were 76.22 ± 27.19 and 76.39 ± 22.52 ng/mL, respectively. However, no significant differences were found in mean C_max_ at week 1 (74.62 ± 49.85 vs. 77.03 ± 13.66; *p* = 0.941) or on average (82.71 ± 42.58 vs. 73.23 ± 12.32; *p* = 0.548) between dogs with and without tumor recurrence (Figure 2).

For the exposure–safety analysis, C_max_ at week 1 and on average and C_min_ were used to compare dogs with and without AE occurrence. Of the nine dogs in which C_max_ was measured, five (55.6%) experienced AEs. Although the difference was not statistically significant, dogs with AEs had a higher mean C_max_ at week 1 (89.32 ± 26.27 vs. 59.85 ± 20.18; *p* = 0.190) and on average (86.20 ± 28.56 vs. 64.14 ± 8.58; *p* = 0.109) than those without AEs. Of the four dogs in which C_min_ was measured, two (50%) experienced AEs, and the overall mean C_min_ was 14.20 ± 9.57 ng/mL. No significant differences in mean C_min_ were found between dogs with and without AEs (13.52 ± 12.05 vs. 14.88 ± 11.29; *p* = 0.667) (Figure 3).

## 4. Discussion

Initially, in human oncology, TKIs were administered orally and at a fixed dose. However, various TKIs exhibit considerable interpatient variability in PK exposure parameters, including C_max_, C_min_, and area under the concentration–time curve, which are associated with efficacy or toxicity [18]. PK exposure can be affected by multiple factors, including differences in gastrointestinal absorption and metabolic processing, interactions with other medications or food, the effects of genetic variation on enzyme activity and transport mechanisms, and organ function impairment [18]. To address interpatient PK variability, dose individualization based on PK exposure, also known as TDM, is essential to avoid unnecessary toxicity and suboptimal efficacy in individual patients. The suitability of TKIs as candidates for TDM depends on their satisfying several criteria, including long-term administration, large variability in PK exposure, narrow therapeutic range, and having defined exposure–response relationships [19].

In veterinary oncology, TOC and sorafenib are the only TKIs for which PK exposure has been studied in dogs with spontaneous tumors [15,20]. Dogs with mast cell tumors that received multiple fixed doses of TOC exhibited high interpatient PK variability. Furthermore, preclinical trials in rodent models have suggested that TOC has a relatively narrow therapeutic range of 50–100 ng/mL [1]. Nonetheless, despite the need for TDM for TOC, the correlations between its TOC exposure and efficacy and safety remain unclear.

This prospective clinical study evaluated the exposure–response relationship of TOC in dogs with solid tumors and explored its feasibility for TDM. In our study, the TOC doses varied among the dogs, with typical doses of 2.4–2.9 mg/kg, which are considered sufficient to achieve a durable clinical benefit [10]. Therefore, we normalized the plasma concentrations to an MTD of 3.25 mg/kg, given that TOC concentration is proportional to dose [15]. Furthermore, we used the plasma concentration measured 6 h after administration as the C_max_. This was based on prior pharmacodynamic findings indicating that C_max_ was reached in 69% of dogs at 6 or 8 h post-administration, with the remaining dogs achieving plasma concentrations >75% of the actual value at that time [10].

The mean interpatient variabilities that we observed in dose-normalized C_max_ and C_min_ (29% and 61%, respectively) are comparable with those reported previously [15]. Interestingly, we did not detect significant differences in dose-normalized C_max_ among weeks 1, 4, and 12. Similarly, in a previous PK study, dose-normalized C_max_ did not significantly differ between days 0 and 86 [15]. For drugs administered at regular intervals, steady-state concentrations are typically achieved after 4–5 half-lives [19]. As its elimination half-life in dogs is approximately 17 h [14], TOC is presumed to reach a steady state within 1 week. The observed consistency in dose-normalized C_max_ after 1 week of TOC treatment is therefore plausible.

Considering the time required for TOC to reach a steady state, we used the PK exposure estimates obtained after 1 week of TOC treatment in the exposure–efficacy and exposure–safety analyses. However, PK exposure and efficacy were not significantly correlated. Furthermore, we were unable to test whether a C_max_ ≥ 40 ng/mL was associated with clinical efficacy in these dogs with solid tumors, because all except one achieved a C_max_ above this threshold during the study. Although statistical significance was not reached in the exposure–safety analysis, a positive trend was observed between AE incidence and C_max_.

Our study had some limitations. Various types of canine tumors were included in the analysis, and differences in behavior among these tumors may have influenced our results. Furthermore, the relatively short follow-up time may not have been sufficient for the development of tumor recurrence, and the small sample size may have increased the risk of type 1 statistical errors. Nevertheless, our results suggest that dose modification based on plasma levels is necessary to prevent AEs during TOC treatment of dogs with solid tumors. This study provides valuable preliminary evidence of the TOC exposure–response relationship in dogs with solid tumors, supporting the feasibility of TOC for TDM.

## 5. Conclusions

To the best of our knowledge, this is the first study to evaluate the exposure–response relationships for TOC. These findings, which reveal a positive association between C_max_ and AE risk, suggest that careful PK monitoring is necessary to mitigate toxicity in dogs treated with TOC, although further large-scale validation studies are warranted.

## Figures and Tables

**Figure 1 animals-15-01025-f001:**
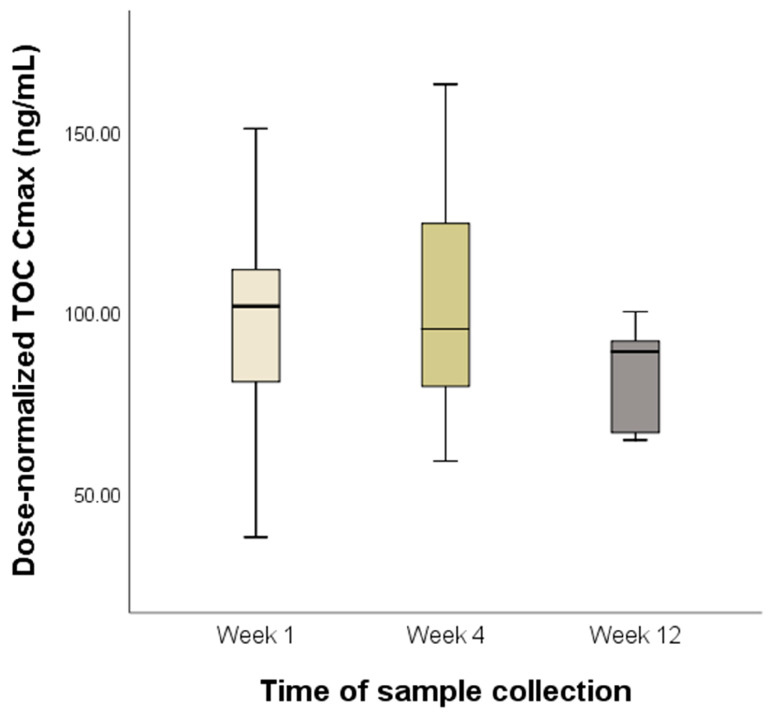
Dose-normalized peak plasma toceranib concentration (C_max_) according to the time of sample collection No significant differences in mean dose-normalized C_max_ were found among weeks 1, 4, and 12 (*p* = 0.414). The boxes extend from the 25th to the 75th percentile, and the horizontal line in the box represents the median. The whiskers indicate the range.

**Figure 2 animals-15-01025-f002:**
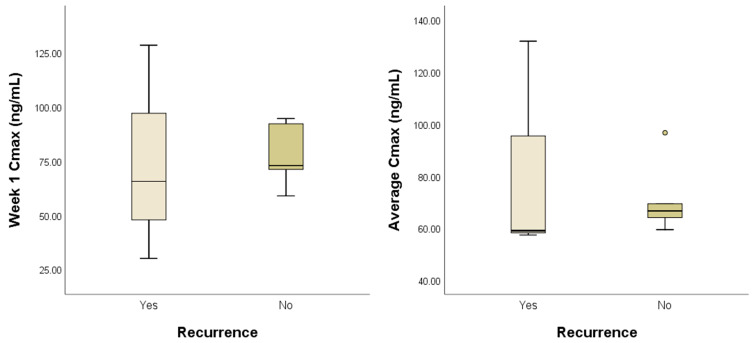
Peak plasma toceranib concentration (C_max_) in dogs with and without tumor recurrence. The groups did not exhibit significant differences in mean C_max_ at week 1 (*p* = 0.941) or on average (*p* = 0.548). The boxes extend from the 25th to the 75th percentile, and the horizontal line in the box represents the median. The whiskers indicate the range, and the circles represent outliers.

**Figure 3 animals-15-01025-f003:**
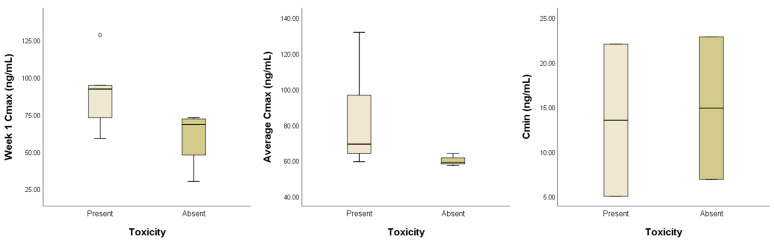
Peak (C_max_) and trough (C_min_) plasma toceranib concentrations in dogs with and without adverse events. The groups did not differ significantly in mean C_max_ (week 1, *p* = 0.190; average, *p* = 0.109) or C_min_ (*p* = 0.667). The boxes extend from the 25th to the 75th percentile, and the horizontal line in the box represents the median. The whiskers indicate the range, and the circles represent outliers.

**Table 1 animals-15-01025-t001:** Patient characteristics.

Case No.	Breed	Age (Years)	Sex	Tumor Location	Diagnosis	TNM
1	Mixed	9	CM	Liver	HCC	T1NXM0
2	Scottish terrier	8	CM	Liver	HCC	T1N0M0
3	Bichon Frisé	10	SF	Liver	HCC	T1N0M0
4	Pomeranian	5	SF	Oral cavity	SCC	T1N1M0
5	Chihuahua	12	SF	Mammary gland	MC	T2N0M0
6	Boston terrier	11	CM	Joint	STS	T2NXM0
7	Coton de Tulear	7	CM	Lung	HS	T1N0M0
8	Japanese spitz	9	CM	Lung	PC	T1N1M0
9	Poodle	11	CM	Liver	HCC	T1N0M0
10	Yorkshire terrier	12	SF	Oral cavity	OMM	T2N1M0

TNM classification was evaluated according to the World Health Organization (WHO) criteria for tumors in domestic animals. TNM, tumor–node–metastasis; CM, castrated male; SF, spayed female; HCC, hepatocellular carcinoma; HS, histiocytic sarcoma; MC, mammary carcinoma; OMM, oral malignant melanoma; PC, pulmonary carcinoma; SCC, squamous cell carcinoma; STS, soft tissue sarcoma.

**Table 2 animals-15-01025-t002:** Information on dogs included in the exposure–efficacy and exposure–safety analyses.

Case No.	Dose (mg/kg)	TOC Duration (d)	C_max_ (ng/mL)			C_min_ (ng/mL)	Recurrence	Toxicity
			Week 1	Week 4	Week 12			
1	2.78	172	72.84	60.1	75.26	-	N	Y
2	2.56	323	72.85	-	79.70	-	N	N
3	2.42	374	58.83	67.59	65.57	<5	N	Y
4	2.77	143	128.38	135.35	-	-	Y	Y
5	2.07	31	71.07	56.96	-	6.89	N	N
6	2.57	36	-	-	-	22.86	N	N
7	2.65	196	92.06	67.74	47.44	22.04	N	Y
8	2.59	130	29.94	87.99	-	-	Y	N
9	2.78	227	94.51	134.61	60.78	-	N	Y
10	2.45	185	65.53	49.07	-	-	Y	N

C_max_, peak concentration; C_min_, trough concentration; TOC, toceranib phosphate; N, no; Y, yes.

## Data Availability

The datasets used and/or analyzed for this study are available from the corresponding author upon reasonable request.

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
