# Peer review of "Exposure–Response Relationships for Toceranib in Dogs with Solid Tumors: A Pilot Study"

_animals, 2025, doi:10.3390/ani15071025_

Round 1
Reviewer 1 Report (Previous Reviewer 3)
Comments and Suggestions for Authors
Exposure-response relationships for toceranib in dogs with 2 solid tumors: A pilot study
Suggestions and comments
Pag 2, line 42
TOC is also effective against various types of canine solid tumors, with a clinical benefit rate of 50–90%
R: can you be more specific of what is the clinical benefit ?
Pag 2, line 68
trough plasma concentration (Cmin)
R: please rephrase trough plasma concentration (Cmin) to minimum plasma concentration
Pag 2 , line 80-81
R: This clinical trial was conducted on client-owned dogs diagnosed with malignant 80 solid tumors between July 2019 and October 2023 at the Konkuk University Animal Can-81 cer Center. But on the paragraph down in the text “ Dogs enrolled between May 2022 and October 2023 underwent blood sample collection to measure plasma TOC concentrations. The period is completely different, can you clarify it
Pag 8, line 253
In the present study, TOC doses varied among dogs, with typical doses of 2.4–2.9 253 mg/kg, which is considered sufficient to achieve a durable clinical benefit
R: Please clarify what is clinical benefit
Table 2
I don´t understand why dog number 6 is still included in the study because only one parameter was estimated (Cmin), can you clarify it
Author Response
|
1) Line 52 : Can you be more specific of what is the clinical benefit ?
|
|
Response: We thank the reviewer for their comments. Based on the studies referenced in this sentence, the dogs were defined as experiencing clinical benefit (CB) if they had a complete response (CR), partial response (PR), or stable disease (SD) (London C et al. 2012. Preliminary evidence for biologic activity of toceranib phosphate (Palladia(®)) in solid tumours. Vet Comp Oncol. 10(3):194-205).
|
|
2) Line 68 : Please rephrase trough plasma concentration (Cmin) to minimum plasma concentration.
|
|
Response: For toceranib phosphate, the trough concentration—measured just before the next dose—represents the lowest concentration observed in the dosing interval. In a pharmacokinetics study on toceranib phosphate, Cmin was defined as the trough concentration (Yancey MF et al. 2012. Pharmacokinetic properties of toceranib phosphate (Palladia, SU11654), a novel tyrosine kinase inhibitor, in laboratory dogs and dogs with mast cell tumors. J Vet Pharmacol Ther. 33(2):162-71). Therefore, we have retained the original terminology.
Line 80-81 : But on the paragraph down in the text “ Dogs enrolled between May 2022 and October 2023 underwent blood sample collection to measure plasma TOC concentrations. The period is completely different, can you clarify it
|
|
Response: We have corrected this sentence as follows: Line 86: “…solid tumors between May 2022 and October 2023 at the Konkuk University …”
|
|
Line 253 : Please clarify what is clinical benefit
|
|
Response: The clinical benefit is defined as the sum of all dogs achieving CR, PR, or SD. This has now been clarified in the revised text.
|
|
Table 2: I don´t understand why dog number 6 is still included in the study because only one parameter was estimated (Cmin), can you clarify it
|
|
Response: Blood samples for Cmax were collected 6 hours after TOC administration into EDTA tubes. The owner of dog 6 was reluctant to wait wait for a long time; therefore we only collected a blood sample for Cmin. |

Reviewer 2 Report (New Reviewer)
Comments and Suggestions for Authors
Veterinary oncology is a rapidly developing field. The research undertaken is fully justified. Due to the number of patients, as the authors themselves noted, they are of a pilot nature.The diversity of tumors in dogs that have undergone therapy is wide and does not allow for binding conclusions.
The goal of the work is clear. The work method is correct. The tables and charts are easy to read.
Statistical analysis is appropriate The applications are appropriate and the writing is correct.
109- Please specify if blood was drawn from the external jugular vein.
Comments on the Quality of English LanguageLanguage allows for the free reading of work
Author Response
|
1) Line 109: Please specify if blood was drawn from the external jugular vein.
|
|
Response: Thank you for this comment. We have revised the sentence as follows:
|
Line 114: “the external jugular vein after starting postoperative…”

Reviewer 3 Report (New Reviewer)
Comments and Suggestions for Authors
General comments
With the evident increase in the lifespan of pets, the incidence of tumors in these animals is becoming more and more common. Therefore the use of oncological drugs for their treatment. Therefore, I consider this manuscript proposal innovative and appropriate for this journal. However, a weakness of this proposal is the lack of a hypothesis to help give more weight to its objective.
Response:
Another possible weakness is the small number of animals, although, it is clear in your title that it is stated that this is a pilot study. I suggest that the small number of animals could be justified as a background for a more elaborate study.
Response:
Finally, the authors suggest an innovative perspective on the application of clinical pharmacokinetics in oncology, however, the justification shown is weak, because the final transition makes it difficult to demonstrate why it is necessary to perform plasma concentration monitoring of toceranib in dogs.
Response:
Particular comments
Lines 13- 16. These introductory lines are too much for a simple summary, if the authors allow me I suggest that lines 13 and 14 be deleted, to be more in line with their objective.
Response:
Line 18. Could the authors please indicate the species and what type of tumors were evaluated?
Response:
Line 19. Please, before describing your results, mention the number of animals, average age, average weight, and general tumor types you evaluated.
Response:
Line 25. Please replace the word “Unnecessary toxicity” with “adverse reactions”.
Response:
Line 28. In complement to my comment made on line 18, I suggest that this same objective should be described in your simple summary to maintain continuity in your idea.
Response:
Line 29. Could you please indicate the average weight and age of the animals?
Response:
Line 40. If the authors agree I suggest that you consider the word “oncology” within your keywords to increase your chances of matching in the different databases.
Response:
Line 45. Please add a reference.
Response:
Line 52. The term “solid tumor” may be a bit ambiguous, if the authors agree they could specify what type of tumors they are referring to, e.g. osteosarcomas or mammary adenocarcinoma.
Response:
Lines 55- 63. This paragraph shows a clear conflict because it does not position the state of the art on the use of toceranib in dogs. First, because the transition of the paragraph is incidental and presents a jump of idea with the previous paragraph. Therefore, I suggest that the authors consider positioning the reader as to what is the main conflict, which according to what has been explained is to establish the effective dose that represents the lowest risk for the appearance of adverse effects. As a second point, I suggest that they offer more detail on previous research such as on the dose where they show greater efficacy of toceranib, but, above all in what type of tumors, since I consider that the greatest innovation of their study is the use of this drug in tumors of other types.
Response:
Line 64. Again, and by what was mentioned in my previous comment, the transition of the text is weak, which weakens its justification, therefore, if the authors allow me, I suggest that they show what evidence exists in human medicine on the dose and plasma concentration necessary to demonstrate the efficacy of toceranib. That can be related to what is described in line 70.
Response:
Lines 67- 69. This idea seems isolated, if the authors agree I suggest that it be deleted.
Response:
Line 76. If the authors agree I suggest adding a hypothesis to reinforce their goal.
Response:
Line 78. In general, I agree with their methodology, however, it is not clear what the study variables were and the general characteristics of their study. If the authors allow me, to make it clear to the reader, I suggest that you add a section called experimental design where you include the evaluation times and the study variables, also, it is not clear if there was division by groups according to their dose, I suggest that this section could help the reader a lot.
Response:
Line 80. Could you specify the general characteristics of your study? That is, whether it was a clinical or experimental study, prospective or retrospective, blinded or double-blinded.
Response:
Lines 82 - 87. This comment is in line with what was mentioned above in line 80. I suggest that you mention the number of animals, the general demographic characteristics of the animals, and, above all, what is confusing if your main interest was to establish what dose allows an effective concentration with less risk of adverse effects. Therefore, I suggest that they clearly describe whether the dose if the dose was for each animal. Also, were the inclusion criteria for the animals?
Response:
Line 115. Could you please specify which analytical equipment was used to perform the measurements of the pharmacological parameters? Also, was it performed by a pharmacologist or a laboratory technician?
Response:
Lines 174- 175. In complement to my comment made on lines 82 - 87 I suggest that this information be relocated earlier.
Response:
Line 177. Previously in your abstract, this parameter was written as a Cmax and Cmin, please homogenize your writing throughout the manuscript.
Response:
Lines 195- 207. From a strictly statistical point of view, it is not clear how a correlation between plasma concentration and the presence of adverse effects was performed. Since a correlation reflects the strength of the relationship between two variables and is plotted linearly or the r value is shown. So I hope the authors understand that this result would be a general conflict for me as it could be considered a serious methodological error. So I suggest that this analysis be done to show it properly or failing that change the name of this variable.
Response:
Line 218 Please correct the spelling error.
Response:
Lines 225- 227. This idea seems to me weak for the beginning of your discussion, if the authors allow me I suggest that you describe what was the most relevant result in your study that can open your discussion.
Response:
Lines 228- 230. This idea is repetitive with what was mentioned in if introduction, I suggest you summarize it to give continuity with your subsequent idea.
Response:
Line 239. This idea is particularly interesting, but could you briefly explain why a difference in interpatient efficacy? For example, could it be primarily due to body conformation in different breeds of dogs? Or even variations in the size and weight of the animals? If the authors will allow me perhaps I would suggest that they briefly discuss whether toceranib exhibits zero order kinetics or order 1 kinetics.
Response:
Line 271. Please, to discuss this I suggest that you can answer my comment made above to your results.
Response:
Line 281. I agree with your limitations, but could you briefly discuss what prospects you would present for study in this area.
Response:
Author Response
|
Comments : With the evident increase in the lifespan of pets, the incidence of tumors in these animals is becoming more and more common. Therefore the use of oncological drugs for their treatment. Therefore, I consider this manuscript proposal innovative and appropriate for this journal. However, a weakness of this proposal is the lack of a hypothesis to help give more weight to its objective.
|
|
Response: Thank you for your comments. We have revised the sentence as follows: Line 79-81: “Given this context, the purpose of this study was to evaluate correlations between TOC exposure and its clinical efficacy and safety and to explore the feasibility of TDM for optimizing TOC treatment in dogs with solid tumors.”
|
|
Comments : Another possible weakness is the small number of animals, although, it is clear in your title that it is stated that this is a pilot study. I suggest that the small number of animals could be justified as a background for a more elaborate study.
|
|
Response: We have revised the sentence as follows to justify our study as a foundation for further research. Line 35-38: “Although further large-scale studies are required, these findings suggest the value of pharmacokinetic monitoring in optimizing TOC dosage and reducing its adverse effects in dogs with solid tumors. Clinicians should consider plasma TOC when managing TOC treatment in small-animal practice.” Line 272-275: “Nevertheless, our results suggest that dose modification based on plasma levels is necessary to prevent AEs during TOC treatment of dogs with solid tumors. This study provides valuable preliminary evidence of the TOC exposure–response relationship in dogs with solid tumors, supporting the feasibility of TOC for TDM.”
Comments : Finally, the authors suggest an innovative perspective on the application of clinical pharmacokinetics in oncology, however, the justification shown is weak, because the final transition makes it difficult to demonstrate why it is necessary to perform plasma concentration monitoring of toceranib in dogs.
|
|
Response: As with the use of TKIs in human medicine, TOC exhibited large interpatient variability in plasma exposure in a previous PK study. However, the exposure–response relationships for TOC efficacy and safety in dogs remains unclear. Therefore, we have added the following sentence to support the necessity of PK monitoring of TOC in dogs. Line 68-75: “For sunitinib, a structural analog of TOC, dose individualization has been proposed for patients with metastatic renal cell tumors, owing to its large interpatient PK variability (34–60%), and TDM-guided dose modification of sunitinib achieved better outcomes while preventing toxicity [13, 14]. Similarly, for TOC, large interpatient variability in plasma exposure was observed in a previous PK study, with interpatient coefficients of variation of 35–40% for Cmax and 30–61% for Cmin..”
|
|
Lines 13- 16 : These introductory lines are too much for a simple summary, if the authors allow me I suggest that lines 13 and 14 be deleted, to be more in line with their objective.
|
|
Response: Thank you for this suggestion. However, we believe that these lines are required to clarify the necessity for therapeutic dose monitoring in dogs receiving toceranib.
|
|
Line 18 : Could the authors please indicate the species and what type of tumors were evaluated?
|
|
Response: We have revised the sentence as follows (further details about the types of tumors are provided in the main text): Line 18: “toceranib phosphate concentration and efficacy and safety in dogs with solid tumors.” |
|
|
|
Line 19 : Please, before describing your results, mention the number of animals, average age, average weight, and general tumor types you evaluated.
|
|
Response: We have chosen not to include these details here, as we feel that they are not required in the Simple Summary. |
|
Line 25 : Please replace the word “Unnecessary toxicity” with “adverse reactions”
|
|
Response: We originally intended ‘unnecessary toxicity’ to refer to adverse events that occur even when a drug is administered at below the maximum tolerated dose. However, to ensure clarity, we have now used, “toxicity at doses below the maximum tolerated dose.” (Line 25). |
|
Line 28 : In complement to my comment made on line 18, I suggest that this same objective should be described in your simple summary to maintain continuity in your idea.
|
|
Response: We have already mentioned this objective in the Simple Summary; therefore, we have not revised this further. We will gladly revise the text further if we have not fully addressed this comment. The term ‘TOC exposure’ has been used interchangeably with plasma TOC concentration. Line 17-19: “…we evaluated correlations between plasma toceranib phosphate concentrations and its efficacy and safety in dogs with solid tumors and explored its feasibility for TDM.” Line 27-28: “Correlations between TOC exposure and efficacy and safety were evaluated in dogs with solid tumors.” |
|
Line 29 : Could you please indicate the average weight and age of the animals?
|
|
Response: The value of including this information in the abstract is not entirely clear to us, so we have not included it here. Additionally, in the other papers published in this journal that we have examined, such information does not seem to be included in the Abstract. If we have not fully understood the reviewer’s comment, we would appreciate further clarification, and we will address it accordingly. |
|
Line 40 : . If the authors agree I suggest that you consider the word “oncology” within your keywords to increase your chances of matching in the different databases.
|
|
Response: We have added the word ‘oncology’ to the list of keywords. |
|
Line 45 : Please add a reference.
|
|
Response: We have added the reference at the end of the sentence. Line 43-45: “Toceranib phosphate (TOC), an orally available small-molecule tyrosine kinase inhibitor (TKI), is the only targeted therapy for dogs approved by the Food and Drug Administration [1].” |
|
Line 52 : The term “solid tumor” may be a bit ambiguous, if the authors agree they could specify what type of tumors they are referring to, e.g. osteosarcomas or mammary adenocarcinoma.
|
|
Response: The term ‘solid tumors’ encompasses all tumors, excluding hematologic malignancies, for which toceranib exhibits clinical benefits. To maintain the flow of the context, we have not mentioned the specific types of tumors (such as anal sac adenocarcinomas, osteosarcomas, thyroid carcinomas, head and neck carcinomas, or nasal carcinomas). |
|
Lines 55- 63 : This paragraph shows a clear conflict because it does not position the state of the art on the use of toceranib in dogs. First, because the transition of the paragraph is incidental and presents a jump of idea with the previous paragraph. Therefore, I suggest that the authors consider positioning the reader as to what is the main conflict, which according to what has been explained is to establish the effective dose that represents the lowest risk for the appearance of adverse effects. As a second point, I suggest that they offer more detail on previous research such as on the dose where they show greater efficacy of toceranib, but, above all in what type of tumors, since I consider that the greatest innovation of their study is the use of this drug in tumors of other types.
|
|
Response: While we respect the reviewer’s opinion, we feel that the logical flow of this paragraph is appropriate, and have not revised it, for several reasons. The paragraph follows the description of toceranib and its efficacy in various types of solid tumors. Initially, TOC was administered at the maximum tolerated dose (MTD), but, importantly, similar benefits were observed at lower doses, with reduced adverse effects. This then leads into the next paragraph, which discusses whether toceranib is appropriate for therapeutic dose monitoring. |
|
Line 64 : Again, and by what was mentioned in my previous comment, the transition of the text is weak, which weakens its justification, therefore, if the authors allow me, I suggest that they show what evidence exists in human medicine on the dose and plasma concentration necessary to demonstrate the efficacy of toceranib. That can be related to what is described in line 70.
|
|
Response: As the reviewer points out, the evidence on dosage and plasma concentration associated with the clinical efficacy of TOC was provided in line 70 (line 73 in the revised manuscript). Furthermore, since toceranib is approved as a TKI only in veterinary practice, there have been no reports of its use in human medicine. Instead, we have added a sentence regarding sunitinib, a structural analog of toceranib, in human medicine. The text has been revised as follows: Lines 68-71: “For sunitinib, a structural analog of TOC, dose individualization has been proposed for patients with metastatic renal cell tumors, owing to its large interpatient PK variability (34–60%), and TDM-guided dose modification of sunitinib achieved better outcomes while preventing toxicity [13, 14]” Lines 332-337: “13. Zhu X, Zhang X, Sun G, Liu Z, Zhang H, Yang Y, Ni Y, Dai J, Zhu S, Chen J, Zhao J, Wang Z, Zeng H, Shen P. Efficacy and Safety of Individualized Schedule of Sunitinib by Drug Monitoring in Patients with Metastatic Renal Cell Carcinoma. Cancer Manag Res. 2021 31(13):6833-6845. doi: 10.2147/CMAR.S327029.” 14. Westerdijk K, Desar IME, Steeghs N, van der Graaf WTA, van Erp NP; Dutch Pharmacology and Oncology Group (DPOG). Imatinib, sunitinib and pazopanib: From flat-fixed dosing towards a pharmacokinetically guided personalized dose. Br J Clin Pharmacol. 2020 86(2):258-273. doi: 10.1111/bcp.14185.”
|
|
Lines 67- 69 : . This idea seems isolated, if the authors agree I suggest that it be deleted.
|
|
Response: We have deleted the following sentence: “Different PK parameters can serve as targets for TDM, such as the peak plasma concentration (Cmax), trough plasma concentration (Cmin), and area under the concentration–time curve” |
|
Line 76 : If the authors agree I suggest adding a hypothesis to reinforce their goal.
|
|
Response: We have revised the sentence as follows. Line 79-81: “Given this context, the purpose of this study was to evaluate correlations between TOC exposure and its clinical efficacy and safety and to explore the feasibility of TDM for optimizing TOC treatment in dogs with solid tumors.” |
|
Line 80 : Could you specify the general characteristics of your study? That is, whether it was a clinical or experimental study, prospective or retrospective, blinded or double-blinded.
|
|
Response: We have revised the sentence as follows: Line 85: “A prospective, randomized, clinical trial was conducted…” |
|
Lines 82 - 87 : This comment is in line with what was mentioned above in line 80. I suggest that you mention the number of animals, the general demographic characteristics of the animals, and, above all, what is confusing if your main interest was to establish what dose allows an effective concentration with less risk of adverse effects. Therefore, I suggest that they clearly describe whether the dose if the dose was for each animal. Also, were the inclusion criteria for the animals?
|
|
Response: Owing to the prospective nature of our study, it would be not appropriate to definitively describe this information in the Materials and Methods section. Rather, such an approach would be more suitable for a retrospective study. Instead, we have explained the study period, the data collected from the enrolled dogs, and the drug regimen in the Materials and Methods section. The details requested by the reviewer have been provided in the Results section. In terms of the inclusion criteria, the study included dogs with solid tumors that were treated with toceranib phosphate, and the details about each dog are presented in the manuscript. If we have not fully understood this comment, we would appreciate further clarification, and we will address it accordingly. Line 155-164 & Table 1: These describe the number of animals and the characteristics of each dog. Table 2: This describes the dose administered to each dog. |
|
Line 115 : Could you please specify which analytical equipment was used to perform the measurements of the pharmacological parameters? Also, was it performed by a pharmacologist or a laboratory technician?
|
|
Response: This information is provided in the Material and Methods section (2.4. Measurement of plasma toceranib concentration). Plasma toceranib was quantified using high-performance liquid chromatography with tandem mass spectrometry detection (LC–MS/MS), using the ExionLC™ AD system (Sciex, Toronto, ON, Canada) and a Triple Quad™ 5500+ mass system (Sciex). Measurements were performed by a pharmacologist. |
|
Lines 174- 175 : In complement to my comment made on lines 82 - 87 I suggest that this information be relocated earlier.
|
|
Response: As we have mentioned in an earlier response, in a prospective study, only the predetermined inclusion criteria at the time of study design are included in the Materials and Methods section, while the actual number of enrolled subjects is typically reported in the Results section. If we have not fully understood this comment, we would appreciate further clarification, and we will address it accordingly. |
|
Line 177 : Previously in your abstract, this parameter was written as a Cmax and Cmin, please homogenize your writing throughout the manuscript.
|
|
Response: We have standardized the terminology to “Cmax” and “Cmin” throughout the manuscript. |
|
Lines 195- 207 : From a strictly statistical point of view, it is not clear how a correlation between plasma concentration and the presence of adverse effects was performed. Since a correlation reflects the strength of the relationship between two variables and is plotted linearly or the r value is shown. So I hope the authors understand that this result would be a general conflict for me as it could be considered a serious methodological error. So I suggest that this analysis be done to show it properly or failing that change the name of this variable.
|
|
Response: Group comparison methods (e.g., t-test or Mann–Whitney U test) are more appropriate than linear correlation analysis when comparing the presence or absence of adverse events, and linear analysis is used to examine relationship between continuous variables (e.g., body weight). However, “correlation” has been used to describe group comparisons in previous exposure–response analyses in human medicine (such as in Teranishi R et al. 2023. Plasma trough concentration of imatinib and its effect on therapeutic efficacy and adverse events in Japanese patients with GIST. Int J Clin Oncol. 28(5):680-687 and Li QB et al. 2010. Imatinib plasma trough concentration and its correlation with characteristics and response in Chinese CML patients. Acta Pharmacol Sin 31(8):999-1004). We have therefore used it throughout the study. Further, we have revised the relevant figure legends to clarify the meaning: Line 204: “Peak plasma toceranib concentration (Cmax) in dogs with and without tumor recurrence” Line 219-220: “Peak (Cmax) and trough (Cmin) plasma toceranib concentrations in dogs with and without adverse events”
|
|
Line 218 : Please correct the spelling error.
|
|
Response: We have corrected the spelling in Figure 3 to “Toxicity.” |
|
Lines 225- 227 : This idea seems to me weak for the beginning of your discussion, if the authors allow me I suggest that you describe what was the most relevant result in your study that can open your discussion.
|
|
Response: As you have has pointed out, the original opening of the Discussion was unnecessary given the objective of the study. Therefore, we have moved that paragraph (original lines 225-235) to later in the text. The Discussion now starts by outlining the necessity of and criteria for therapeutic drug monitoring and explaining why toceranib is a suitable candidate, by examining how it satisfies these needs and criteria. |
|
Line 239 : This idea is particularly interesting, but could you briefly explain why a difference in interpatient efficacy? For example, could it be primarily due to body conformation in different breeds of dogs? Or even variations in the size and weight of the animals? If the authors will allow me perhaps I would suggest that they briefly discuss whether toceranib exhibits zero order kinetics or order 1 kinetics.
|
|
Response: Many factors account for this variability, including breed, body weight, differences in absorption, genetic polymorphisms in metabolizing enzymes, and interactions with food and co-medications. We now discuss the causes of this interpatient variability in plasma levels as follows: Line 227-230: “PK exposure can be affected by multiple factors, including differences in gastrointestinal absorption and metabolic processing, interactions with other medications or food, the effects of genetic variation on enzyme activity and transport mechanisms, and organ function impairment.” While the inability to control these variables is a limitation of our study, our primary focus was to evaluate correlations between plasma TOC concentration and recurrence or AEs, aiming to account for these individual differences. Toceranib phosphate follows first-order kinetics, as it is eliminated dose-proportionally, maintains a consistent half-life, and exhibits linear pharmacokinetics without evidence of saturation leading to zero-order kinetics (Yancey MF et al. 2012. Pharmacokinetic properties of toceranib phosphate (Palladia, SU11654), a novel tyrosine kinase inhibitor, in laboratory dogs and dogs with mast cell tumors. J Vet Pharmacol Ther. 33(2):162-71). |
|
Line 271 : Please, to discuss this I suggest that you can answer my comment made above to your results.
|
|
Response: As we have mentioned above, “correlation” has been used for group comparison in previous exposure–response analyses in human medicine. If we have not fully understood this comment, we would appreciate further clarification, and we will address it accordingly. |
|
Line 281 : I agree with your limitations, but could you briefly discuss what prospects you would present for study in this area.
|
|
Response: We have added the following sentence to address this: Lines 272-275: “Nevertheless, our results suggest that dose modification based on plasma levels is necessary to prevent AEs during TOC treatment of dogs with solid tumors. This study provides valuable preliminary evidence of the TOC exposure–response relationship in dogs with solid tumors, supporting the feasibility of TOC for TDM” |

Reviewer 4 Report (New Reviewer)
Comments and Suggestions for Authors
The manuscript presents a valuable contribution to veterinary oncology by examining the pharmacokinetic variability and exposure-response relationships of Toceranib phosphate (TOC) in dogs with solid tumors. TOC is widely used in veterinary oncology, yet its pharmacokinetic variability and exposure-response relationship remain poorly understood. This study provides findings in drug monitoring to optimize efficacy and reduce adverse events and contributes to veterinary oncology by examining the pharmacokinetic variability and exposure-response relationships of Toceranib phosphate (TOC) in dogs with solid tumors.
The authors' experimental approach is well-structured and interpatient variability in TOC exposure. Statistical approaches are appropriate for the dataset and provide solid data for interpretation. Although not statistically significant (it can be improved with the larger sample size), the observed trend of higher Cmax being associated with increased AEs supports the potential need for dose adjustments in veterinary patients.
Their findings will benefit the community, and I recommend publishing this manuscript in Animals after these considerations.
- Some places are highlighted. Please check and remove or change the words.
- Error bars are too wide for week 1 and 4 in Figure 1.
- Also, in Figure 2, while a trend toward increased AEs at higher Cmax levels was observed, it doesn't seem the correlation between TOC exposure and tumor recurrence is statistically significant. This issue could be due to the small sample size, interpatient variability, or confounding factors such as tumor heterogeneity, and it could be improved with the following suggestions.
- How do you manage the interpatient differences in breed, body weight, and metabolism? These could contribute to the observed pharmacokinetic variability. Also, some dogs received concomitant medications, which may have influenced TOC metabolism or AE occurrence.
Author Response
|
Comments : Some places are highlighted. Please check and remove or change the words. Error bars are too wide for week 1 and 4 in Figure 1.
|
|
Response: Thank you for these comments. We have removed the highlighting in the text. As described in the Figure 1 legend, the error bars (whiskers) indicate the range of dose-normalized TOC Cmax at each week. The whiskers appear wide because of the broad range of Cmax values.
|
|
Comments : How do you manage the interpatient differences in breed, body weight, and metabolism? These could contribute to the observed pharmacokinetic variability. Also, some dogs received concomitant medications, which may have influenced TOC metabolism or AE occurrence.
|
|
Response: As correctly pointed out, several factors, including breed, body weight, and metabolism, can influence drug pharmacokinetics. While the inability to control these variables is a limitation of our study, our primary focus was to evaluate the correlations between plasma TOC concentration and recurrence or AEs, aiming to account for these individual differences. Although we did not control for these factors, our study was intended for preliminary exploration, and its findings warrant further validation through additional experiments using a larger cohort of dogs. |

Round 2
Reviewer 3 Report (New Reviewer)
Comments and Suggestions for Authors
General comments
I appreciate the authors for considering most of the comments made above in their manuscript. However, I see with sadness the refusal of some recommendations made to their proposal that I consider should be made because they involve serious errors and would possibly lead to misinterpretation.
Response:
Particular comments
Simple summary:
I understand the justification for keeping the current state of your simple summary, however, according to your comments it may be contrasting to keep your introductory lines and not clarify the methodology of your study briefly. Which from a reader's perspective may be confusing and may not be of interest in a manuscript that I consider contributes a lot. Therefore, I reiterate to the authors my comments made earlier which I suggest they reconsider.
Response:
Lines 13- 16. These introductory lines are too much for a simple summary, if the authors allow me I suggest that lines 13 and 14 be deleted, to be more in line with their objective.
Response:
Line 18. Could the authors please indicate the species and what type of tumors were evaluated?
Response:
Line 19. Please, before describing your results, mention the number of animals, average age, average weight, and general tumor types you evaluated.
Response:
Line 25. I understand your explanation, but, strictly speaking, the definition mentioned refers to “side effects” or “adverse reactions”. Mentioning that there is “unnecessary toxicity” could lead to thinking if there is any necessary toxicity. Please, I invite the authors to rethink if this terminology is correct.
Response:
Line 29. I understand and it might not explain my point well, but, I suggest including the average age and weight of the animals because this might give a general idea if it is a clinical study or a highly controlled experimental study, for a reader who only has access to the abstract.
Response:
Line 195. I understand the justification of the authors and I share the view on the analysis of the results, however, the fact that the authors mention in their line 194 that they presented a correlation between Cmax and tumor recurrence can be misunderstood from a statistical point of view. Although I understand that this term has been used in the relationship between two variables, this is wrong, because I suggest that they should justify from a statistical point of view that they only evaluated the differences between the groups or if they evaluated the level of linear relationship between two variables in this case Cmax and tumor recurrence. Please understand that this can be a serious confusion for the reader.
Response:
Line 238. Again, as I explained earlier in the results, mentioning that there is a correlation means that there is a strong or weak, inverse or direct relationship between two variables. From what I understand they are using it as a way of referring to the differences found in their groups, which as I have mentioned can be a serious interpretation error, please, I suggest that they modify it and that the wording of this can be changed. Or else I suggest that you perform the analysis of the correlation between these two variables which would possibly add more value to your study.
Response:
Author Response
|
Comments : I understand the justification for keeping the current state of your simple summary, however, according to your comments it may be contrasting to keep your introductory lines and not clarify the methodology of your study briefly. Which from a reader's perspective may be confusing and may not be of interest in a manuscript that I consider contributes a lot. Therefore, I reiterate to the authors my comments made earlier which I suggest they reconsider.
|
|
Response: We thank the reviewer for comments. We appreciate the constructive feedback provided during the revision, which has helped improve the manuscript. We have revised the sentence to reflect the reviewer’s suggestions as much as possible in second round while maintaining the integrity of our study.
|
|
Lines 13- 16 : These introductory lines are too much for a simple summary, if the authors allow me I suggest that lines 13 and 14 be deleted, to be more in line with their objective.
|
|
Response: We thank the reviewer for comments. We have deleted the following sentence. Line 13-16: “Given the exposure–response relationships (regarding efficacy/toxicity) for several tyrosine kinase inhibitors, therapeutic drug monitoring (TDM) has been proposed to optimize their efficacy and reduce toxicity.”
Line 18 : Could the authors please indicate the species and what type of tumors were evaluated?
|
|
Response: We have added the sentence as follows: Line 17-20: “In this study, we included 10 dogs with 7 solid tumors, including hepatocellular carcinoma (n = 4), squamous cell carcinoma (n = 1), soft tissue sarcoma (n = 1), histiocytic sarcoma (n = 1), oral malignant melanoma (n = 1), mammary carcinoma (n = 1), and pulmonary carcinoma (n = 1).”
|
|
Line 19 : Please, before describing your results, mention the number of animals, average age, average weight, and general tumor types you evaluated.
|
|
Response: We thank the reviewer for comments. We have added a sentence providing information about the number of animals and tumor types, along with their frequencies (line 17-20). However, we are uncertain whether including the average age and average weight is essential in the ‘simple summary’ and ‘abstract’ section in our study. Furthermore, adding this information would exceed the word limits specified for each section according to instruction guide. Additionally, we found no similar precedents in articles recently published in Animals. Therefore, to ensure clarity and conciseness in the overall context, we have opted not to include this information. We kindly ask the reviewer to reconsider this suggestion.
|
|
Line 25 : I understand your explanation, but, strictly speaking, the definition mentioned refers to “side effects” or “adverse reactions”. Mentioning that there is “unnecessary toxicity” could lead to thinking if there is any necessary toxicity. Please, I invite the authors to rethink if this terminology is correct.
|
|
Response: We thank the reviewer for comments. We have replaced the term ‘unnecessary toxicity’ with ‘adverse events’ to maintain consistency throughout the manuscript. Line 26-27: “…dose adjustment to avoid potential adverse events and suboptimal efficacy in targeted therapy.”
|
|
Line 29 : I understand and it might not explain my point well, but, I suggest including the average age and weight of the animals because this might give a general idea if it is a clinical study or a highly controlled experimental study, for a reader who only has access to the abstract.
|
|
Response: We thank the reviewer for comments. Unfortunately, if we added this information, abstract would exceed 200 words limit. As mentioned above, these factors are not directly related to the study’s outcomes. Therefore, we kindly ask the reviewer to reconsider this suggestion.
|
|
Line 195 : I understand the justification of the authors and I share the view on the analysis of the results, however, the fact that the authors mention in their line 194 that they presented a correlation between Cmax and tumor recurrence can be misunderstood from a statistical point of view. Although I understand that this term has been used in the relationship between two variables, this is wrong, because I suggest that they should justify from a statistical point of view that they only evaluated the differences between the groups or if they evaluated the level of linear relationship between two variables in this case Cmax and tumor recurrence. Please understand that this can be a serious confusion for the reader.
|
|
Response: We thank the reviewer for comments. As the reviewer suggested, we have revised the sentence as follows. Line 194-195: “For the exposure–efficacy analysis, Cmax values at week 1 and on average were used to compare dog with and without tumor recurrence.” Line 207-208: “For the exposure–safety analysis, Cmax at week 1 and on average and Cmin were used to compare dogs with and without AE occurrence.”
|
|
Line 238 : Again, as I explained earlier in the results, mentioning that there is a correlation means that there is a strong or weak, inverse or direct relationship between two variables. From what I understand they are using it as a way of referring to the differences found in their groups, which as I have mentioned can be a serious interpretation error, please, I suggest that they modify it and that the wording of this can be changed. Or else I suggest that you perform the analysis of the correlation between these two variables which would possibly add more value to your study
|
|
Response: We thank the reviewer for comments. Our study primarily aimed to investigate the relationship between TOC exposure and efficacy or safety. However, given the lack of difference in plasma TOC concentration between the groups, assessing a linear relationship was not feasible, which we acknowledge as a limitation of our study. Therefore, we would like to retain the term 'correlation' when describing the objective of our study (line 15, 28, 239). However, if the reviewer prefers a more broad term such as 'relationship' or 'association,' we are willing to revise it accordingly.
|

Reviewer 4 Report (New Reviewer)
Comments and Suggestions for Authors
I pointed out some weaknesses in the first round, and the authors added comments and revised the manuscript. It could be accepted as preliminary research, as authors mentioned.
Author Response
|
Comments: I pointed out some weaknesses in the first round, and the authors added comments and revised the manuscript. It could be accepted as preliminary research, as authors mentioned.
|
|
Response: We thank the reviewer for their comments. We appreciate the constructive feedback provided in the first round, which has helped improve the manuscript. As noted, this study serves as preliminary research, and we hope it contributes valuable insights to the field. |

Round 3
Reviewer 3 Report (New Reviewer)
Comments and Suggestions for Authors
I appreciate the authors' consideration of the changes suggested above and understand that this may be a limitation from a statistical point of view. I am only suggesting minimal formatting changes.
Particular comments
Please add the space between references to be consistent with the rest of your citations.
This manuscript is a resubmission of an earlier submission. The following is a list of the peer review reports and author responses from that submission.
Round 1
Reviewer 1 Report
Comments and Suggestions for Authors
The manuscript has several flaws that really limitate the scientific conclusions that can be taken from the study, e.g.:
- sample size is very limited (n=10);
- the tumors are very different;
- the study is very limitated in time, doesn't allowing to following the patients and their responses.
Author Response
Reviwer 1
|
1. The manuscript has several flaws that really limitate the scientific conclusions that can be taken from the study, e.g.: sample size is very limited (n=10); the tumors are very different; the study is very limitated in time, doesn't allowing to following the patients and their responses.
|
|
Response: We thank the reviewer for the comment. As reviewer rightly pointed out, there were several limitations in our study, including sample size, tumor types, and short study period. We have, therefore, described the title of the manuscript to explicitly state that this is a “pilot study”. This title is aimed at emphasizing the fact that the sample size used by us was intended for preliminary exploration and the findings of the study warrant further validation through additional experiments with a larger cohort of dogs. We understand the importance of conducting a robust statistical analysis, including correlation analysis, to draw more definitive conclusions, and would ensure that these aspects are thoroughly addressed in future studies. |
Reviewer 2 Report
Comments and Suggestions for Authors
The evaluation of exposure-response relationships for efficacy and toxicity is very important nowadays, considering the expansion of the use of TKI. Although the small sample size and heterogenous tumors used in the manuscript, the research showed the potential possibility of increase risk of AE for higher Cmax. Some issues are point out below and in the document attached.
Line 109. Blood sampling: Whole blood was collected after starting postoperative TOC treatment, When was it? posoperative would be after surgery, when?
In table 1 (line 165) Some description in the legend do not correspond properly.
Figure 3. In the figure of Week 1 Cmax, in the text (line 208) authors said "dogs with AEs had a higher mean Cmax at week 1 (89.32  26.27 vs. 59.85  20.18; p = 0.190)" The absent column of toxicity, if you observe the number 59.85 looks like do not correspond in the "y axis" with the mean and standard deviation. The same for the Average Cmax in present toxicity. Please check it. Or maybe I am wrong.

Author Response
Reviewer 2
|
1. Line 109: Blood sampling: Whole blood was collected after starting postoperative TOC treatment, When was it? posoperative would be after surgery, when?
|
|
Response: We thank the reviewer for the comment. TOC treatment was started 1 month after surgery. We have add the information about median time interval from surgery to postoperative TOC treatment as follows.
Line 162-164: “ The median time interval from surgery to postoperative TOC treatment was 29 days (range, 14–47).”
|
|
2. Line 165: In table 1, some description in the legend do not correspond properly.
|
|
Response: We thank the reviewer for the comment. We have corrected description of table 1 legend in line 169-171.
|
|
3. Line 208: In the figure 3 of Week 1 Cmax, in the text, authors said "dogs with AEs had a higher mean Cmax at week 1 (89.32 ± 26.27 vs. 59.85 ± 20.18; p = 0.190)" The absent column of toxicity, if you observe the number 59.85 looks like do not correspond in the "y axis" with the mean and standard deviation. The same for the Average Cmax in present toxicity. Please check it. Or maybe I am wrong.
|
|
Response: We thank the reviewer for the comment. Box and whisker plot consists of 1st and 3rd quartiles and the median, which is different from the mean value. Therefore, we have added information in legend of all figures as follows.
Each box represents the 25th and 75th percentiles and the horizontal line in the box represents the median. The whiskers indicate the range.
|
Reviewer 3 Report
Comments and Suggestions for Authors
Dear Authors
General comments
The theme of these research paper is very interesting, the methodologies are sound and adequate however I think there is a foundation error that influences the result. The sample size calculation was not done and by consequence the power statistics of the sample were not estimated which is essential to this type of study. The power statistics estimation is a step to avoid rejecting a null hypothesis that is actually true in the population (type I error). Can you please provide this information!
Pag 3, line 106
TOC was permanently discontinued upon request of the owner or if unacceptable AEs occurred
Commments: what kind of unacceptable AEs are listed and classified as unacceptable
Pag 2, line 81
Comments: In the materials and methods section, sub-section animals, please state the inclusion and exclusion criteria used for this study
Pag3, line 119
Toceranib was quantitated
Comments: replace by toceranib was quantified
Pag 3, line 175
During the follow-up period, 3 of 10 dogs (30%) developed local recurrence, and 5 of 10 (50%) experienced adverse events.
Comments: please provide the list of adverse events observed
Pag 3, line 114
The plasma was separated, divided into cryovials, and stored at -80 ℃ until analysis
Comments. If the plasma was stored at -80ºC it was possible to increase the sample number by performing this study in a multicenter setting.
Pag 8, line 272
Various types of canine tumors were included in the analysis, and differences in behavior among tumors may have influenced our results.
Comments: That is variable that should be studied, especially in dogs with hepatic tumours, because Toceranib is mainly metabolized in liver. So, one variable that should be studied is the influence of tumor location in adverse events occurrence
Other important variable to consider is plasma protein (namely albumin) binding of toceranib. According to previous studies binding of Toceranib in fresh plasma protein ranged from 90.8% to 92.8% at concentrations between 20 ng/mL and 500 ng/mL. For that reason, in the presence of hypoalbuminemia, the ative metabolite can be more available to cause adverse events, the concentration of albumin should be studied in Exposure-response relationships of adverse events
Author Response
Reviewer 3
|
1. Line 106: what kind of unacceptable AEs are listed and classified as unacceptable
|
|
Response: We thank the reviewer for the comment. In case no. 1, grade 3 diarrhea occurred even after the discontinuation and re-administration of TOC. Therefore, we had no choice but to permanently discontinue TOC treatment.
|
|
2. Line 81: In the materials and methods section, sub-section animals, please state the inclusion and exclusion criteria used for this study.
|
|
Response: We thank the reviewer for the comment. We included all dogs diagnosed with malignant solid tumors in our study and also mentioned the exclusion criteria in Animals section (Line 88-90).
|
|
3. Line 119: replace by toceranib was quantified
|
|
Response: We thank the reviewer for the comment. We have corrected sentence as follows.
Line 117: “Toceranib was quantified in plasma samples…”
|
|
4. Line 175: please provide the list of adverse events observed
|
|
Response: We thank the reviewer for the comment. We have added information about adverse events in Table S1.
Line 177: “…5 of 10 (50%) experienced adverse events (Table S1)”
|
|
5. Line 115: If the plasma was stored at -80ºC, it was possible to increase the sample number by performing this study in a multicenter setting.
|
|
Response: We thank the reviewer for the comment. Unfortunately, it was difficult to increase the sample size because the supply of TOC in our country was not stable during the study period. Nevertheless, our study would serve as a foundation for further large-scale studies on the exposure-response relationship for toceranib in dogs with solid tumors.
|
|
6. Line 272: That is variable that should be studied, especially in dogs with hepatic tumours, because Toceranib is mainly metabolized in liver. So, one variable that should be studied is the influence of tumor location in adverse events occurrence
|
|
Response: We thank the reviewer for the comment. As reviewer proposed, we analyze the influence of tumor location in adverse events occurrence; however, there was no significance in AE occurrence according to tumor location in regression analysis (p = 1.000).
|
|
7. Line 272: Other important variable to consider is plasma protein (namely albumin) binding of toceranib. According to previous studies binding of Toceranib in fresh plasma protein ranged from 90.8% to 92.8% at concentrations between 20 ng/mL and 500 ng/mL. For that reason, in the presence of hypoalbuminemia, the ative metabolite can be more available to cause adverse events, the concentration of albumin should be studied in Exposure-response relationships of adverse events
|
|
Response: We thank the reviewer for the comment. All dogs enrolled in our study showed serum albumin concentration within normal range (2.2-3.9 mg/dL).
|